# Relational State-Space Model for Stochastic Multi-Object Systems

**Fan Yang[†], Ling Chen[*†], Fan Zhou[†], Yusong Gao[‡], Wei Cao[‡]**

[†]College of Computer Science and Technology, Zhejiang University, Hangzhou, China

[‡]Alibaba Group, Hangzhou, China

`{fanyang01,lingchen,fanzhou}@zju.edu.cn`
`{jianchuan.gys,mingsong.cw}@alibaba-inc.com`

## Abstract

Real-world dynamical systems often consist of multiple stochastic subsystems that interact with each other. Modeling and forecasting the behavior of such dynamics are generally not easy, due to the inherent hardness in understanding the complicated interactions and evolutions of their constituents. This paper introduces the relational state-space model (R-SSM), a sequential hierarchical latent variable model that makes use of graph neural networks (GNNs) to simulate the joint state transitions of multiple correlated objects. By letting GNNs cooperate with SSM, R-SSM provides a flexible way to incorporate relational information into the modeling of multi-object dynamics. We further suggest augmenting the model with normalizing flows instantiated for vertex-indexed random variables and propose two auxiliary contrastive objectives to facilitate the learning. The utility of R-SSM is empirically evaluated on synthetic and real time series datasets.

## 1 Introduction

Many real-world dynamical systems can be decomposed into smaller interacting subsystems if we take a fine-grained view. For example, the trajectories of coupled particles are co-determined by per-particle physical properties (e.g., mass and velocity) and their physical interactions (e.g., gravity); traffic flow can be viewed as the coevolution of a large number of vehicle dynamics. Models that are able to better capture the complex behavior of such multi-object systems are of wide interest to various communities, e.g., physics, ecology, biology, geoscience, and finance.

State-space models (SSMs) are a wide class of sequential latent variable models (LVMs) that serve as workhorses for the analysis of dynamical systems and sequence data. Although SSMs are traditionally designed under the guidance of domain-specific knowledge or tractability consideration, recently introduced deep SSMs (Fraccaro, 2018) use neural networks (NNs) to parameterize flexible state transitions and emissions, achieving much higher expressivity. To develop deep SSMs for multi-object systems, graph neural networks (GNNs) emerge to be a promising choice, as they have been shown to be fundamental NN building blocks that can impose *relational inductive bias* explicitly and model complex interactions effectively (Battaglia et al., 2018).

Recent works that advocate GNNs for modeling multi-object dynamics mostly make use of GNNs in an autoregressive (AR) fashion. AR models based on recurrent (G)NNs can be viewed as special instantiations of SSMs in which the state transitions are restricted to being deterministic (Fraccaro, 2018, Section 4.2). Despite their simplicity, it has been pointed out that their modeling capability is bottlenecked by the deterministic state transitions (Chung et al., 2015; Fraccaro et al., 2016) and the oversimplified observation distributions (Yang et al., 2018).

In this study, we make the following contributions: **(i)** We propose the relational state-space model (R-SSM), a novel hierarchical deep SSM that simulates the stochastic state transitions of interacting objects with GNNs, extending GNN-based dynamics modeling to challenging stochastic multi-object systems. **(ii)** We suggest using the graph normalizing flow (GNF) to construct expressive joint

---

[*]Corresponding author.

state distributions for R-SSM, further enhancing its ability to capture the joint evolutions of corre-lated stochastic subsystems. **(iii)** We develop structured posterior approximation to learn R-SSM using variational inference and introduce two auxiliary training objectives to facilitate the learning.

Our experiments on synthetic and real-world time series datasets show that R-SSM achieves com-petitive test likelihood and good prediction performance in comparison to GNN-based AR models and other sequential LVMs. The remainder of this paper is organized as follows: Section 2 briefly reviews neccesary preliminaries. Section 3 introduces R-SSM formally and presents the methods to learn R-SSM from observations. Related work is summarized in Section 4 and experimental evaluation is presented in Section 5. We conclude the paper in Section 6.

## 2 PRELIMINARIES

In this work, an attributed directed graph is given by a 4-tuple: $\mathcal{G} = (\mathcal{V}, \mathcal{E}, \mathbf{V}, \mathbf{E})$, where $\mathcal{V} = [N] := \{1, \ldots, N\}$ is the set of vertices, $\mathcal{E} \subseteq [N] \times [N]$ is the set of edges, $\mathbf{V} \in \mathbb{R}^{N \times d_v}$ is a matrix of static vertex attributes, and $\mathbf{E} \in \mathbb{R}^{N \times N \times d_e}$ is a sparse tensor storing the static edge attributes. The set of direct predecessors of vertex $i$ is notated as $\mathcal{N}_i^- = \{p | (p, i) \in \mathcal{E}\}$. We use the notation $\mathbf{x}_i$ to refer to the $i$-th row of matrix $\mathbf{X}$ and write $\mathbf{x}_{ij}$ to indicate the $(i, j)$-th entry of tensor $\mathbf{X}$ (if the corresponding matrix or tensor appears in the context). For sequences, we write $\mathbf{x}_{\leq t} = \mathbf{x}_{1:t} := (\mathbf{x}_1, \ldots, \mathbf{x}_t)$ and switch to $\mathbf{x}_t^{(i)}$ for referring to the $i$-th row of matrix $\mathbf{X}_t$.

### 2.1 GRAPH NEURAL NETWORKS

GNNs are a class of neural networks developed to process graph-structured data and support rela-tional reasoning. Here we focus on vertex-centric GNNs that iteratively update the vertex repre-sentations of a graph $\mathcal{G}$ while being *equivariant* (Maron et al., 2019) under vertex relabeling. Let $\mathbf{H} \in \mathbb{R}^{N \times d}$ be a matrix of vertex representations, in which the $i$-th row $\mathbf{h}_i \in \mathbb{R}^d$ is the vectorized representation attached to vertex $i$. Conditioning on the static graph structure and attributes given by $\mathcal{G}$, a GNN just takes the vertex representations $\mathbf{H}$ along with some graph-level context $\mathbf{g} \in \mathbb{R}^{d_g}$ as input and returns new vertex representations $\mathbf{H}' \in \mathbb{R}^{N \times d'}$ as output, i.e., $\mathbf{H}' = \text{GNN}(\mathcal{G}, \mathbf{g}, \mathbf{H})$.

When updating the representation of vertex $i$ from $\mathbf{h}_i$ to $\mathbf{h}_i'$, a GNN takes the representations of other nearby vertices into consideration. Popular GNN variants achieve this through a multi-round message passing paradigm, in which the vertices repeatedly send messages to their neighbors, ag-gregate the messeages they received, and update their own representations accordingly. Formally, the operations performed by a basic block of a message-passing GNN are defined as follows:

$$\forall (j, i) \in \mathcal{E}: \qquad \mathcal{M}_{j \to i} = \text{MESSAGE}\left(\mathbf{g}, \mathbf{v}_j, \mathbf{v}_i, \mathbf{e}_{ji}, \mathbf{h}_j, \mathbf{h}_i\right) \qquad (1)$$

$$\forall i \in \mathcal{V}: \qquad \mathcal{A}_i = \text{AGGREGATE}\left(\{\mathcal{M}_{p \to i}\}_{p \in \mathcal{N}_i^-}\right) \qquad (2)$$

$$\forall i \in \mathcal{V}: \qquad \mathbf{h}_i' = \text{COMBINE}\left(\mathbf{g}, \mathbf{v}_i, \mathbf{h}_i, \mathcal{A}_i\right) \qquad (3)$$

Throughout this work, we implement Equations (1) and (2) by adopting a multi-head attention mech-anism similar to Vaswani et al. (2017) and Velikovi et al. (2018). For Equation (3), we use either a RNN cell or a residual block (He et al., 2016), depending on whether the inputs to GNN are RNN states or not. We write such a block as $\mathbf{H}' = \text{MHA}(\mathcal{G}, \mathbf{g}, \mathbf{H})$ and give its detailed implementation in the Appendix. A GNN simply stacks $L$ separately-parameterized MHA blocks and iteratively computes $\mathbf{H} =: \mathbf{H}^{(0)}, \ldots, \mathbf{H}^{(L)} =: \mathbf{H}'$, in which $\mathbf{H}^{(l)} = \text{MHA}(\mathcal{G}, \mathbf{g}, \mathbf{H}^{(l-1)})$ for $l = 1, \ldots, L$. We write this construction as $\mathbf{H}' = \text{GNN}(\mathcal{G}, \mathbf{g}, \mathbf{H})$ and treat it as a black box to avoid notational clutter.

### 2.2 STATE-SPACE MODELS

State-space models are widely applied to analyze dynamical systems whose true states are not directly observable. Formally, an SSM assumes the dynamical system follows a latent state pro-cess $\{\mathbf{z}_t\}_{t \geq 1}$, which possibly depends on exogenous inputs $\{\mathbf{u}_t\}_{t \geq 1}$. Parameterized by some (unknown) static parameter $\theta$, the latent state process is characterized by an initial density $\mathbf{z}_1 \sim \pi_\theta(\cdot | \mathbf{u}_1)$ and a transition density $\mathbf{z}_{t+1} \sim f_\theta(\cdot | \mathbf{z}_{\leq t}, \mathbf{u}_{\leq t+1})$. Moreover, at each time step, some noisy measurements of the latent state are observed through an observation density:

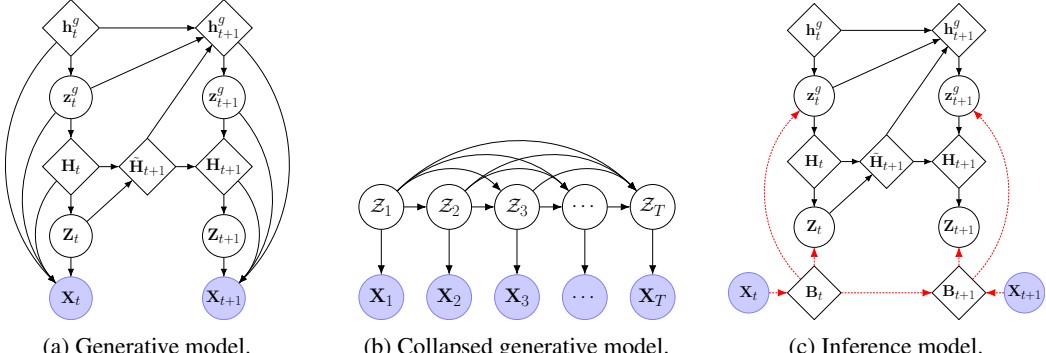

(a) Generative model.    (b) Collapsed generative model.    (c) Inference model.

Figure 1: Graphical structures of R-SSM. Diamonds represent deterministic states and circles represent random variables. To be concise, the dependencies on the graph $\mathcal{G}$ and exogenous inputs $\mathbf{U}_{1:T}$ are not shown. (b) is the result of collapsing all deterministic states in (a) and writing $\mathcal{Z}_t = (\mathbf{z}_t^g, \mathbf{Z}_t)$. In (c), solid lines represent the computation shared with the generative model and dashed lines represent additional computation for inference.

$\mathbf{x}_t \sim g_\theta\left(\cdot | \mathbf{z}_{\leq t}, \mathbf{u}_{\leq t}\right)$. The joint density of $\mathbf{x}_{1:T}$ and $\mathbf{z}_{1:T}$ factors as: $p(\mathbf{x}_{1:T}, \mathbf{z}_{1:T} | \mathbf{u}_{1:T}) = \pi_\theta(\mathbf{z}_1 | \mathbf{u}_1) \prod_{t=2}^{T} f_\theta(\mathbf{z}_t | \mathbf{z}_{<t}, \mathbf{u}_{\leq t}) \prod_{t=1}^{T} g_\theta(\mathbf{x}_t | \mathbf{z}_{\leq t}, \mathbf{u}_{\leq t})$.

The superior expressiveness of SSMs can be seen from the fact that the marginal predictive distribution $p(\mathbf{x}_t | \mathbf{x}_{<t}, \mathbf{u}_{\leq t}) = \int p(\mathbf{x}_t, \mathbf{z}_{\leq t} | \mathbf{x}_{<t}, \mathbf{u}_{\leq t}) \, \mathrm{d}\mathbf{z}_{\leq t}$ can be far more complex than unimodal distributions and their finite mixtures that are common in AR models. Recently developed deep SSMs use RNNs to compress $\mathbf{z}_{\leq t}$ (and $\mathbf{u}_{\leq t}$) into fixed-size vectors to achieve tractability. As shown in next section, R-SSM can be viewed as enabling multiple individual deep SSMs to communicate.

## 2.3 Normalizing Flows

Normalizing flows (Rezende & Mohamed, 2015) are invertible transformations that have the capability to transform a simple probability density into a complex one (or vice versa). Given two domains $\mathcal{X} \subseteq \mathbb{R}^D$ and $\mathcal{Y} \subseteq \mathbb{R}^D$, let $f : \mathcal{X} \to \mathcal{Y}$ be an invertible mapping with inverse $f^{-1}$. Applying $f$ to a random variable $\mathbf{z} \in \mathcal{X}$ with density $p(\mathbf{z})$, by the *change of variables* rule, the resulting random variable $\mathbf{z}' = f(\mathbf{z}) \in \mathcal{Y}$ will have a density:

$$p(\mathbf{z}') = p(\mathbf{z}) \left| \det \frac{\partial f^{-1}}{\partial \mathbf{z}'} \right| = p(\mathbf{z}) \left| \det \frac{\partial f}{\partial \mathbf{z}} \right|^{-1}$$

A series of invertible mappings with cheap-to-evaluate determinants can be chained together to achieve complex transformations while retaining efficient density calculation. This provides a powerful way to construct expressive distributions.

## 3 Relational State-Space Model

Suppose there is a dynamical system that consists of multiple interacting objects, and observing this system at a specific time is accomplished by acquiring measurements from every individual object simultaneously. We further assume these objects are homogeneous, i.e., they share the same measurement model, and leave systems whose constituents are nonhomogeneous for future work. To generatively model a time-ordered series of observations collected from this system, the straightforward approach that builds an individual SSM for each object is usually unsatisfactory, as it simply assumes the state of each object evolves independently and ignores the interactions between objects. To break such an independence assumption, our main idea is to let multiple individual SSMs interact through GNNs, which are expected to capture the joint state transitions of correlated objects well.

## 3.1 GENERATIVE MODEL

Given the observations for a multi-object dynamical system, our model further assumes its inter-action structure is known as prior knowledge. The interaction structure is provided as a directed graph, in which each object corresponds to a vertex, and a directed edge indicates that the state of its head is likely to be affected by its tail. In situations where such graph structure is not available, a complete graph can be specified. However, to model dynamical systems comprising a large number of objects, it is often beneficial to explicitly specify sparse graph structures, because they impose stronger relational inductive bias and help save the computational cost.

A relational state-space model assumes a set of correlated dynamical subsystems evolve jointly under the coordination of graph neural networks. Formally, given a graph $\mathcal{G} = (\mathcal{V}, \mathcal{E}, \mathbf{V}, \mathbf{E})$, in which an edge $(i, j) \in \mathcal{E}$ indicates that the state of vertex $j$ may be affected by vertex $i$. Let $\mathbf{u}_t^{(i)} \in \mathbb{R}^{d_u}$ and $\mathbf{x}_t^{(i)} \in \mathbb{R}^{d_x}$ be the input and observation for vertex $i$ at time step $t$, respectively. For $T$ steps, we introduce a set of unobserved random variables $\{\mathbf{z}_{1:T}^{(i)}\}_{i=1}^N$, in which $\mathbf{z}_t^{(i)} \in \mathbb{R}^{d_z}$ represents the latent state of vertex $i$ at time step $t$. Furthermore, we introduce a global latent variable $\mathbf{z}_t^g \in \mathbb{R}^{d_g}$ for each time step to represent the global state shared by all vertices. Conditioning on the graph and exogenous inputs, an R-SSM factorizes the joint density of observations and latent states as follows:

$$p_\theta \left( \{\mathbf{x}_{1:T}^{(i)}\}_{i=1}^N, \{\mathbf{z}_{1:T}^{(i)}\}_{i=1}^N, \mathbf{z}_{1:T}^g \Big| \mathcal{G}, \{\mathbf{u}_{1:T}^{(i)}\}_{i=1}^N \right) =$$

$$\prod_{t=1}^T f_\theta \left( \{\mathbf{z}_t^{(i)}\}_{i=1}^N, \mathbf{z}_t^g \Big| \{\mathbf{z}_{<t}^{(i)}\}_{i=1}^N, \mathbf{z}_{<t}^g, \mathcal{G}, \{\mathbf{u}_{\leq t}^{(i)}\}_{i=1}^N \right)$$

$$\prod_{t=1}^T \prod_{i=1}^N g_\theta \left( \mathbf{x}_t^{(i)} \Big| \mathbf{z}_t^{(i)}, \mathbf{z}_{\leq t}^g, \{\mathbf{z}_{<t}^{(i)}\}_{i=1}^N, \mathcal{G}, \{\mathbf{u}_{\leq t}^{(i)}\}_{i=1}^N \right) \quad (4)$$

For notational simplicity, we switch to the matrix notation $\mathbf{Z}_t = \left[ \mathbf{z}_t^{(1)}, \mathbf{z}_t^{(2)}, \ldots, \mathbf{z}_t^{(N)} \right]^\top$ from now on. The joint transition density $f_\theta$ is further factorized as a product of global transition density $f_\theta^g$ and local transition density $f_\theta^\star$, i.e., $f_\theta \left( \mathbf{Z}_t, \mathbf{z}_t^g | \ldots \right) = f_\theta^g \left( \mathbf{z}_t^g | \ldots \right) f_\theta^\star \left( \mathbf{Z}_t | \mathbf{z}_t^g, \ldots \right)$. To instantiate these conditional distributions, a GNN accompanied by RNN cells is adopted to recurrently compress the past dependencies at each time step into fixed-size context vectors. Specifically, the observations are assumed to be generated from following process:

$$\tilde{\mathbf{h}}_t^{(i)} = \text{RNN}_\theta^v \left( \mathbf{h}_{t-1}^{(i)}, [\mathbf{z}_{t-1}^{(i)}, \mathbf{u}_t^{(i)}] \right), \quad \tilde{\mathbf{h}}_t^g = \text{READOUT}_\theta \left( \mathcal{G}, \tilde{\mathbf{H}}_t \right)$$

$$\mathbf{h}_t^g = \text{RNN}_\theta^g \left( \mathbf{h}_{t-1}^g, [\mathbf{z}_{t-1}^g, \tilde{\mathbf{h}}_t^g] \right), \quad \mathbf{z}_t^g \sim f_\theta^g \left( \cdot \, | \mathbf{h}_t^g \right)$$

$$\mathbf{H}_t = \text{GNN}_\theta \left( \mathcal{G}, \mathbf{z}_t^g, \tilde{\mathbf{H}}_t \right), \quad \mathbf{Z}_t \sim f_\theta^\star \left( \cdot \, | \mathbf{H}_t \right)$$

$$\mathbf{x}_t^{(i)} \sim g_\theta \left( \cdot \, | \mathbf{z}_t^{(i)}, \mathbf{h}_t^{(i)}, \mathbf{z}_t^g, \mathbf{h}_t^g \right)$$

for $i = 1, \ldots, N$ and $t = 1, \ldots, T$, where $\mathbf{h}_0^{(i)} = \mathbf{h}_0^\star$ and $\mathbf{z}_0^{(i)} = \mathbf{z}_0^\star$. Here $\mathbf{h}_0^g$, $\mathbf{z}_0^g$, $\mathbf{h}_0^\star$ and $\mathbf{z}_0^\star$ are learnable initial states. The READOUT function aggregates the context vectors of all vertices into a global context vector in a permutation-invariant manner. The global transition density $f_\theta^g$ is specified to be a diagonal Gaussian distribution whose mean and variance are parameterized by the output of a multilayer perceptron (MLP), and the local transition density $f_\theta^\star$ will be discussed later. The local observation distribution $g_\theta$ can be freely selected in line with the data, and in our experiments it is either a Gaussian distribution or a mixture of logistic distributions parameterized by MLPs.

The graphical structure of two consecutive steps of the generating process is illustrated in Figure 1a. An intuitive way to think about this generative model is to note that the $N + 1$ latent state processes interact through the GNN, which enables the new state of a vertex to depend on not only its own state trajectory but also the state trajectories of other vertices and the entire graph.

## 3.2 LEARNING AND INFERENCE

As illustrated in Figure 1b, writing $\mathcal{Z}_t = (\mathbf{z}_t^g, \mathbf{Z}_t)$ and suppressing the dependencies on the graph $\mathcal{G}$ and exogenous inputs $\mathbf{U}_{1:T}$, an R-SSM can be interpreted as an ordinary SSM in which the entire graph evolves as a whole, i.e., the joint density of latent states and observations factors as:

$p(\mathbf{X}_{1:T}, \mathcal{Z}_{1:T}) = \prod_{t=1}^{T} p_\theta(\mathcal{Z}_t|\mathcal{Z}_{<t}) p_\theta(\mathbf{X}_t|\mathcal{Z}_t)$. Given observations $\mathbf{X}_{1:T}$, we are interested in learning unknown parameters $\theta$ and inferring unobserved states $\mathcal{Z}_{1:T}$. For the learning task we wish to maximize the marginal likelihood $p_\theta(\mathbf{X}_{1:T}) = \int p_\theta(\mathbf{X}_{1:T}, \mathcal{Z}_{1:T}) \, d\mathcal{Z}_{1:T}$, but in our case the integral is intractable. We adopt a recently developed variational inference (VI) approach called variational sequential Monte Carlo (VSMC) (Maddison et al., 2017; Naesseth et al., 2018; Le et al., 2018), which maximizes a variational lower bound on the log marginal likelihood instead and learns the proposal distributions for the inference task simultaneously.

Given a sequence of proposal distributions $\{q_\phi (\mathcal{Z}_t|\mathcal{Z}_{<t}, \mathbf{X}_{\leq t})\}_{t=1}^{T}$ parameterized by $\phi$, running the sequential Monte Carlo (SMC) algorithm with $K$ particles yields an unbiased marginal likelihood estimator $\hat{p}_{\theta,\phi,K}(\mathbf{X}_{1:T}) = \prod_{t=1}^{T} \left[ 1/K \sum_{k=1}^{K} w_t^k \right]$, where $w_t^k$ is the unnormalized importance weight of particle $k$ at time $t$. The variational lower bound is obtained by applying the Jensen's inequality:

$$\mathcal{L}_K^{\text{SMC}}(\theta, \phi) \coloneqq \mathbb{E}\left[\log \hat{p}_{\theta,\phi,K}(\mathbf{X}_{1:T})\right] \leq \log \mathbb{E}\left[\hat{p}_{\theta,\phi,K}(\mathbf{X}_{1:T})\right] = \log p_\theta(\mathbf{X}_{1:T}) \tag{5}$$

Assuming the proposal distributions are reparameterizable (Kingma & Welling, 2014), we use the biased gradient estimator $\nabla \mathcal{L}_K^{\text{SMC}}(\theta, \phi) \approx \mathbb{E}[\nabla \log \hat{p}_{\theta,\phi,K}(\mathbf{X}_{1:T})]$ to maximize $\mathcal{L}_K^{\text{SMC}}$.

**Proposal design.** We make the proposal for $\mathcal{Z}_t$ depend on the information up to time $t$ and share some parameters with the generative model. We also choose to factorize $q_\phi (\mathcal{Z}_t|\ldots) = r_\phi^g (\mathbf{z}_t^g|\ldots) r_\phi^\star (\mathbf{Z}_t|\mathbf{z}_t^g,\ldots)$. The proposal distributions for all time steps are structured as follows:

$$\tilde{\mathbf{b}}_t^{(i)} = \text{RNN}_\phi(\mathbf{b}_{t-1}^{(i)}, [\mathbf{x}_t^{(i)}, \mathbf{u}_t^{(i)}]), \quad \tilde{\mathbf{b}}_t^g = \text{READOUT}_{\phi,1}(\mathcal{G}, \tilde{\mathbf{B}}_t)$$

$$\mathbf{B}_t = \text{GNN}_\phi(\mathcal{G}, \tilde{\mathbf{b}}_t^g, \tilde{\mathbf{B}}_t), \quad \mathbf{b}_t^g = \text{READOUT}_{\phi,2}(\mathcal{G}, \mathbf{B}_t)$$

$$\mathbf{z}_t^g \sim r_\phi^g(\cdot \,|\mathbf{h}_t^g, \mathbf{b}_t^g), \quad \mathbf{Z}_t \sim r_\phi^\star(\cdot \,|\mathbf{H}_t, \mathbf{B}_t)$$

for $i = 1, \ldots, N$ and $t = 1, \ldots, T$, where $\mathbf{h}_t^g$ and $\mathbf{H}_{1:T}$ are computed using the relevant parts of the generative model. $r_\phi^g$ is specified to be a diagonal Gaussian parameterized by an MLP, and $r_\phi^\star$ will be discussed soon. Here $\mathbf{B}_t$ can be interpreted as a *belief state* (Gregor et al., 2019), which summarizes past observations $\mathbf{X}_{\leq t}$ (and inputs $\mathbf{U}_{\leq t}$) deterministically. The graphical structure of this proposal design is shown in Figure 1c, and the detailed VSMC implementation using this proposal is given in Appendix A.4.

## 3.3 GRAPH NORMALIZING FLOW

The local transition density $f_\theta^\star (\mathbf{Z}_t|\ldots)$ and the local proposal density $r_\phi^\star (\mathbf{Z}_t|\ldots)$ may be parameterized in several ways. One simple and efficient starting point is (block-)diagonal Gaussian distribution: $f_\theta^\star (\mathbf{Z}_t|\ldots) = \prod_{i=1}^{N} \mathcal{N}(\mathbf{z}_t^{(i)}|\ldots)$, which assumes that the object states are conditionally independent, i.e., the joint state distribution is completely factorized over objects. We believe that such an independence assumption is an oversimplification for situations where the joint state evolution is multimodal and highly correlated. One possible way to introduce inter-object dependencies is modeling joint state distributions as Markov random fields (MRFs) (Naesseth et al., 2019), but this will significantly complicate the learning.

Here we introduce the Graph Normalizing Flow (GNF) [1], which adapts Glow (Kingma & Dhariwal, 2018) to graph settings and enables us to build expressive joint distributions for correlated random variables indexed by graph nodes. As described earlier, the key ingredient for a flow is a series invertible mappings that are iteratively applied to the samples of a base distribution. Now we are interested in the case where the samples are vertex states $\mathbf{Z}_t$, and thus the invertible mappings should be further constrained to be *equivariant* under vertex relabeling. This rules out popular autoregressive flows, e.g., IAF (Kingma et al., 2016) and MAF (Papamakarios et al., 2017).

Our GNF is built upon the *coupling* layer introduced in Dinh et al. (2017), which provides a flexible framework to construct efficient invertible mappings. A GNF coupling layer splits the input $\mathbf{Z} \in \mathbb{R}^{N \times D}$ into two parts, $\mathbf{Z}_a \in \mathbb{R}^{N \times d}$ and $\mathbf{Z}_b \in \mathbb{R}^{N \times (D-d)}$. The output $\mathbf{Z}' \in \mathbb{R}^{N \times D}$ is formed as:

$$\mathbf{Z}_a' = \mathbf{Z}_a, \quad \mathbf{Z}_b' = \mathbf{Z}_b \odot \exp\left(s(\mathbf{Z}_a)\right) + t(\mathbf{Z}_a), \quad \mathbf{Z}' = [\mathbf{Z}_a', \mathbf{Z}_b'],$$

---

[1]GNF has been independently developed by Liu et al. (2019) for different purpose.

where $\odot$ denotes the element-wise product, and the functions $s(\cdot)$ and $t(\cdot)$ are specified to be GNNs to enforce the equivariance property. A GNF combines a coupling layer with a trainable element-wise affine layer and an invertible $1 \times 1$ convolution layer (Hoogeboom et al., 2019), organizing them as: Input $\rightarrow$ Affine $\rightarrow$ Coupling $\rightarrow$ Conv$_{1 \times 1}$ $\rightarrow$ Output. A visual illustration of this architecture is provided in Appendix A.5.

In order to obtain more expressive prior and variational posterior approximation, the local transition density and local proposal density can be constructed by stacking multiple GNFs on top of diagonal Gaussian distributions parameterized by MLPs. With the message passing inside the coupling layers, GNFs can transform independent noise into correlated noise and thus increase model expressivity. The $1 \times 1$ convolution layers free us from manually permuting the dimensions, and the element-wise affine layers enable us to tune their initial weights to stablize training.

## 3.4 AUXILIARY CONTRASTIVE PREDICTION TASKS

In our initial experiments, we found that learning R-SSM suffered from the *posterior collpase* phenomenon, which is a well known problem in the training of variational autoencoders (VAEs). It means that the variational posterior approximation $q_\phi(\mathcal{Z}_t | \mathcal{Z}_{<t}, \mathbf{X}_{\le t})$ degenerate into the prior $f_\theta(\mathcal{Z}_t | \mathcal{Z}_{<t})$ in the early stage of optimization, making the training dynamics get stuck in undesirable local optima. Besides, we also encountered a more subtle problem inherent in likelihood-based training of deep sequential models. That is, for relatively smooth observations, the learned model tended to only capture short-term local correlations but not the interaction effects and long-term transition dynamics.

Motivated by recent advances in unsupervised representation learning based on mutual information maximization, in particular the Contrastive Predictive Coding (CPC) approach (Oord et al., 2018), we alleviate these problems by forcing the latent states to perform two auxiliary contrastive prediction tasks. At each time step $t$, the future observations of each vertex $i$ are summarized into a vector using a backward RNN: $\mathbf{c}_t^{(i)} = \text{RNN}_\psi(\mathbf{x}_{>t}^{(i)})$. Then we define two auxiliary CPC objectives:

$$\mathcal{L}_1^{\text{aux}} = \mathbb{E}\left[\sum_{t=1}^{T-1}\sum_{i=1}^{N} \log \frac{\lambda_{\psi,1}(\hat{\mathbf{z}}_t^{(i)}, \mathbf{c}_t^{(i)})}{\sum_{\mathbf{c} \in \Omega_{t,i}} \lambda_{\psi,1}(\hat{\mathbf{z}}_t^{(i)}, \mathbf{c})}\right], \quad \mathcal{L}_2^{\text{aux}} = \mathbb{E}\left[\sum_{t=1}^{T-1}\sum_{i=1}^{N} \log \frac{\lambda_{\psi,2}(\hat{\mathbf{h}}_t^{(i)}, \mathbf{c}_t^{(i)})}{\sum_{\mathbf{c} \in \Omega_{t,i}} \lambda_{\psi,2}(\hat{\mathbf{h}}_t^{(i)}, \mathbf{c})}\right]$$

where $\hat{\mathbf{z}}_t^{(i)} = [\mathbf{z}_t^g, \mathbf{z}_t^{(i)}]$, $\hat{\mathbf{h}}_t^{(i)} = \text{MLP}_\psi(\sum_{j \ne i \,\wedge\, j \in \mathcal{N}_i^-} \mathbf{h}_t^{(j)})$, and $\Omega_{t,i}$ is a set that contains $\mathbf{c}_t^{(i)}$ and some negative samples. The expectation is over negative samples and the latent states sampled from the filtering distributions. The positive score functions $\lambda_{\psi,1}$ and $\lambda_{\psi,2}$ are specified to be simple log-bilinear models.

Intuitively, $\mathcal{L}_1^{\text{aux}}$ encourages the latent states to encode useful information that helps distinguish the future summaries from negative samples. $\mathcal{L}_2^{\text{aux}}$ encourages the deterministic states to reflect the interaction effects, as it contrastingly predicts the future summary of vertex $i$ based on the states of $i$'s neighbors only. The negative samples are selected from the future summaries of other vertices within the minibatch. The final objective to maximize is $\mathcal{L} = \mathcal{L}_K^{\text{SMC}} + \beta_1 \mathcal{L}_1^{\text{aux}} + \beta_2 \mathcal{L}_2^{\text{aux}}$, in which $\beta_1 \ge 0$ and $\beta_2 \ge 0$ are tunable hyperparameters. The procedure to estimate this objective is described in Appendix A.4.

## 4 RELATED WORK

**GNN-based dynamics modeling.** GNNs (Scarselli et al., 2009; Duvenaud et al., 2015; Li et al., 2016; Defferrard et al., 2016; Gilmer et al., 2017; Hamilton et al., 2017; Velikovi et al., 2018; Xu et al., 2019; Maron et al., 2019) provide a promising framework to learn on graph-structured data and impose relational inductive bias in learning models. We refer the reader to Battaglia et al. (2018) for a recent review. GNNs (or neural message passing modules) are the core components of recently developed neural physics simulators (Battaglia et al., 2016; Watters et al., 2017; Chang et al., 2017; Janner et al., 2019; Sanchez-Gonzalez et al., 2018; Mrowca et al., 2018; Li et al., 2019) and spatiotemporal or multi-agent dynamics models (Alahi et al., 2016; Hoshen, 2017; Li et al., 2018; Zhang et al., 2018; Tacchetti et al., 2019; Chen et al., 2020). In these works, GNNs usually act autoregressively or be integrated into the sequence-to-sequence (seq2seq) framework (Sutskever et al.,

2014). Besides, recently they have been combined with generative adversarial networks (Goodfellow et al., 2014) and normalizing flows for multi-agent forecasting (Gupta et al., 2018; Kosaraju et al., 2019; Rhinehart et al., 2019). R-SSM differs from all these works by introducing structured latent variables to represent the uncertainty on state transition and estimation.

**GNNs in sequential LVMs.** A few recent works have combined GNNs with a sequential latent variable model, including R-NEM (van Steenkiste et al., 2018), NRI (Kipf et al., 2018), SQAIR (Kosiorek et al., 2018), VGRNN (Hajiramezanali et al., 2019), MFP (Tang & Salakhutdinov, 2019), and Graph VRNN (Sun et al., 2019; Yeh et al., 2019). The latent variables in R-NEM and NRI are discrete and represent membership relations and types of edges, respectively. In contrast, the latent variables in our model are continuous and represent the states of objects. SQAIR is also a deep SSM for multi-object dynamics, but the GNN is only used in its inference network. VGRNN is focused on modeling the topological evolution of dynamical graphs. MFP employs a conditional VAE architecture, in which the per-agent discrete latent variables are shared by all time steps. The work most relevant to ours is Graph VRNN, in which the hidden states of per-agent VRNNs interact through GNNs. Our work mainly differs from it by introducing a global latent state process to make the model hierarchical and exploring the use of normalizing flows as well as the auxiliary contrastive objectives. More subtle differences are discussed in Section 5.2.

**Deep LVMs for sequential data.** There has been growing interest in developing latent variable models for sequential data with neural networks as their building blocks, among which the works most relevant to ours are stochastic RNNs and deep SSMs. Many works have proposed incorporating stochastic latent variables into vanilla RNNs to equip them with the ability to express more complex data distributions (Bayer & Osendorfer, 2014; Chung et al., 2015; Fraccaro et al., 2016; Goyal et al., 2017; Ke et al., 2019) or, from another perspective, developing deep SSMs by parameterizing flexible transition and emission distributions using neural networks (Krishnan et al., 2017; Fraccaro et al., 2017; Buesing et al., 2018; Zheng et al., 2017; Hafner et al., 2019). Approximate inference and parameter estimation methods for nonlinear SSMs have been extensively studied in the literature (Doucet & Johansen, 2009; Andrieu et al., 2010; Kantas et al., 2015; Gu et al., 2015; Karl et al., 2016; Marino et al., 2018; Gregor et al., 2019; Hirt & Dellaportas, 2019). We choose VSMC (Maddison et al., 2017; Naesseth et al., 2018; Le et al., 2018) as it combines the powers of VI and SMC. The posterior collapse problem is commonly addressed by KL annealing, which does not work with VSMC. The idea of using auxiliary costs to train deep SSMs has been explored in Z-forcing (Goyal et al., 2017; Ke et al., 2019), which predicts the future summaries directly rather than contrastingly. As a result, the backward RNN in Z-forcing may degenerate easily.

## 5 EXPERIMENTS

We implement R-SSM using the TensorFlow Probability library (Dillon et al., 2017). The experiments are organized as follows: In Section 5.1, we sample a toy dataset from a simple stochastic multi-object model and validate that R-SSM can fit it well while AR models and non-relational models may struggle. In Section 5.2, R-SSM is compared with state-of-the-art sequential LVMs for multi-agent modeling on a basketball gameplay dataset, and the effectiveness of GNF is tested through ablation studies. Finally, in Section 5.3, the prediction performance of R-SSM is compared with strong GNN-based seq2seq baselines on a road traffic dataset. Due to the space constraint, the detailed model architecture and hyperparameter settings for each dataset are given in the Appendix. Below, all values reported with error bars are averaged over 3 or 5 runs.

### 5.1 SYNTHETIC TOY DATASET

First we construct a simple toy dataset to illustrate the capability of R-SSM. Each example in this dataset is generated by the following procedure:

$$\mathcal{G} \sim \text{SBM}(N, K, p_0, p_1), \quad \mathbf{v}_i \sim \text{Normal}(\mathbf{0}, \mathbf{I}), \quad z_0^{(i)} \sim \text{Normal}(0, 1)$$
$$\tilde{z}_t^{(i)} = \boldsymbol{\eta}^\top \mathbf{v}_i + \alpha_1 \sum_{j \in \mathcal{N}_i} z_{t-1}^{(j)} / |\mathcal{N}_i| + \alpha_2 z_{t-1}^{(i)} \tag{6}$$
$$z_t^{(i)} \sim \text{Normal}\big(\cos(\tilde{z}_t^{(i)}), \sigma_z^2\big), \quad x_t^{(i)} \sim \text{Normal}\big(\tanh(\varepsilon z_t^{(i)}), \sigma_x^2\big)$$

Table 2: Test log-likelihood and rollout quality comparisons on the basketball gameplay dataset (offensive players only).

| Model | $\mathcal{L}_{1000}^{\text{SMC}}$ | ELBO | Speed | Distance | OOB |
|---|---|---|---|---|---|
| VRNN | — | 2360 | 0.89 | 43.78 | 33.78 |
| MI-VRNN | — | 2362 | 0.79 | 38.92 | 15.52 |
| R-SSM | $2459.8_{\pm.3}$ | $2372.3_{\pm.8}$ | $0.83_{\pm.01}$ | $40.75_{\pm.15}$ | $1.84_{\pm.16}$ |
| $+\mathcal{L}_2^{\text{aux}}$ | $2463.3_{\pm.4}$ | $2380.2_{\pm.6}$ | $0.82_{\pm.01}$ | $40.36_{\pm.23}$ | $2.17_{\pm.09}$ |
| $+$GNF (4) | $2483.2_{\pm.3}$ | $2381.6_{\pm.4}$ | $0.80_{\pm.00}$ | $39.37_{\pm.35}$ | $2.06_{\pm.15}$ |
| $+$GNF (8) | $\mathbf{2501.6}_{\pm.2}$ | $\mathbf{2382.1}_{\pm.4}$ | $0.79_{\pm.00}$ | $39.14_{\pm.29}$ | $2.12_{\pm.10}$ |
| Ground Truth | — | — | 0.77 | 37.78 | 2.21 |

for $i = 1, \ldots, N$ and $t = 1, \ldots, T$. Here SBM is short for the symmetric stochastic block model, in which each vertex $i$ belongs to exact one of the $K$ communities, and two vertices $i$ and $j$ are connected with probability $p_0$ if they are in the same community, $p_1$ otherwise. A vertex-specific covariate vector $\mathbf{v}_i \in \mathbb{R}^{d_v}$ is attached to each vertex $i$, and by Equation (6), the state of each vertex $i$ can be affected by its neighbors $\mathcal{N}_i$. Choosing the parameters $d_v = 4$, $N = 36$, $K = 3$, $p_0 = 1/3$, $p_1 = 1/18$, $T = 80$, $\alpha_1 = 5.0$, $\alpha_2 = -1.5$, $\boldsymbol{\eta} = [-1.5, 0.4, 2.0, -0.9]^\top$, $\sigma_x = \sigma_z = 0.05$, and $\varepsilon = 2.5$, we generate 10K examples for training, validation, and test, respectively. A typical example is visualized in the Appendix.

Despite the simple generating process, the resulting dataset is highly challenging for common models to fit. To show this, we compare R-SSM with several baselines, including (a) VAR: Fitting a first-order vector autoregression model for each example; (b) VRNN: A variational RNN (Chung et al., 2015) shared by all examples; (c) GNN-AR: A variant of the recurrent decoder of NRI (Kipf et al., 2018), which is exactly a GNN-based AR model when given the ground-truth graph. VAR and VRNN are given access to the observations $\{x_{1:T}^{(i)}\}_{i=1}^N$ only, while GNN-AR and R-SSM are additionally given access to the graph structure $(\mathcal{V}, \mathcal{E})$ (but not the vertex covariates). GNF is not used in R-SSM because the true joint transition distribution is factorized over vertices.

For each model, we calculate three metrics: (1) LL: Average log-likelihood (or its lower bound) of test examples; (2) MSE: Average mean squared one-step prediction error given the first 75 time steps of each test example; (3) CP: Average coverage probability of a 90% one-step prediction interval. For non-analytic models, point predictions and prediction intervals are computed using 1000 Monte Carlo samples. The results are reported in Table 1.

Table 1: Test log-likelihood and prediction performance comparisons on the synthetic toy dataset.

| Model | LL | MSE | CP |
|---|---|---|---|
| VAR | -366 | $0.679_{\pm.000}$ | $0.750_{\pm.000}$ |
| VRNN | $\geq$-2641 | $0.501_{\pm.003}$ | $0.931_{\pm.002}$ |
| GNN-AR | -94 | $0.286_{\pm.002}$ | $0.806_{\pm.004}$ |
| R-SSM | $\geq$2583 | $0.029_{\pm.001}$ | $0.883_{\pm.002}$ |
| $+\mathcal{L}_2^{\text{aux}}$ | $\geq\mathbf{2647}$ | $\mathbf{0.024}_{\pm.001}$ | $0.897_{\pm.001}$ |

The generating process involves latent factors and nonlinearities, so VAR performs poorly as expected. VRNN largely underfits the data and struggles to generalize, which may be caused by the different topologies under the examples. In contrast, GNN-AR and R-SSM generalize well as expected, while R-SSM achieves much higher test log-likelihood and produces good one-step probabilistic predictions. This toy case illustrates the generalization ability of GNNs and suggests the importance of latent variables for capturing the uncertainty in stochastic multi-object systems. We also observed that without $\mathcal{L}_1^{\text{aux}}$ the training dynamics easily get stuck in posterior collapse at the very early stage, and adding $\mathcal{L}_2^{\text{aux}}$ help improve the test likelihood.

## 5.2 BASKETBALL GAMEPLAY

In basketball gameplay, the trajectories of players and the ball are highly correlated and demonstrate rich, dynamic interations. Here we compare R-SSM with a state-of-the-art hierarchical sequential LVM for multi-agent trajectories (Zhan et al., 2019), in which the per-agent VRNNs are coordinated

Table 3: Test log-likelihood comparison on the basketball gameplay dataset (offensive players plus the ball).

| Model | LL | $\mathcal{L}_{1000}^{\mathrm{SMC}}$ |
|---|---|---|
| Yeh et al. (2019) | | |
| GRNN | 2264 | — |
| VRNN | >2750 | — |
| GVRNN | >**2832** | — |
| R-SSM+$\mathcal{L}_2^{\mathrm{aux}}$ | >2761 $\pm 1$ | 2805 $\pm 0$ |
| +GNF (8) | >2783 $\pm 1$ | 2826 $\pm 0$ |

Table 4: Forecast MAE comparison on the METR-LA dataset. $h$ is the number of steps predicted into the future. The $\mathbf{X}_{t-h}$ baseline outputs $\mathbf{X}_{t-h}$ to predict $\mathbf{X}_t$.

| Model | $h = 3$ | $h = 6$ | $h = 12$ |
|---|---|---|---|
| $\mathbf{X}_{t-h}$ | 3.97 | 4.99 | 6.65 |
| DCRNN | 2.77 | 3.15 | **3.60** |
| GaAN | 2.71 | **3.12** | 3.64 |
| R-SSM | **2.67** $\pm .00$ | 3.14 $\pm .01$ | 3.72 $\pm .02$ |
| CP | 0.896 $\pm .001$ | 0.891 $\pm .001$ | 0.883 $\pm .002$ |

by a global "macro intent" model. We note it as MI-VRNN. The dataset[2] includes 107,146 training examples and 13,845 test examples, each of which contains the 2D trajectories of ten players and the ball recorded at 6Hz for 50 time steps. Following their settings, we use the trajectories of offensive team only and preprocess the data in exactly the same way to make the results directly comparable. The complete graph of players is used as the input to R-SSM.

Several ablation studies are performed to verify the utility of the proposed ideas. In Table 3, we report test likelihood bounds and the rollout quality evaluated with three heuristic statistics: average speed (feet/step), average distance traveled (feet), and the percentage of out-of-bound (OOB) time steps. The VRNN baseline developed by Zhan et al. (2019) is also included for comparison. Note that the VSMC bound $\mathcal{L}_{1000}^{\mathrm{SMC}}$ is a tighter log-likelihood approximation than the ELBO (which is equivalent to $\mathcal{L}_1^{\mathrm{SMC}}$). The rollout statistics of R-SSMs are calculated from 150K 50-step rollouts with 10 burn-in steps. Several selected rollouts are visualized in the Appendix.

As illustrated in Table 2, all R-SSMs outperform the baselines in terms of average test log-likelihood. Again, we observed that adding $\mathcal{L}_1^{\mathrm{aux}}$ is necessary for training R-SSM successfully on this dataset. Training with the proposed auxiliary loss $\mathcal{L}_2^{\mathrm{aux}}$ and adding GNFs do improve the results. R-SSM with 8 GNFs (4 in prior, 4 in proposal) achieves higher likelihood than R-SSM with 4 GNFs, indicating that increasing the expressivity of joint state distributions helps fit the data better. As for the rollout quality, the OOB rate of the rollouts sampled from our model matches the ground-truth significantly better, while the other two statistics are comparable to the MI-VRNN baseline.

In Table 3, we also provide preliminary results for the setting that additionally includes the trajectory of the ball. This enables us to compare with the results reported by Yeh et al. (2019) for Graph VRNN (GVRNN). The complete graph of ball and players served as input to R-SSM is annotated with two node types (player or ball) and three edge types (player-to-ball, ball-to-player or player-to-player). R-SSM achieves competitive test likelihood, and adding GNFs helps improve the performance. We point out that several noticeable design choices of GVRNN may help it outperform R-SSM: **(i)** GVRNN uses a GNN-based observation model, while R-SSM uses a simple factorized observation model. **(ii)** GVRNN encodes $\mathbf{X}_{1:t-1}$ into $\mathbf{H}_t$ and thus enables the prior of $\mathbf{Z}_t$ to depend on past observations, which is not the case in R-SSM. **(iii)** GVRNN uses several implementation tricks, e.g., predicting the changes in observations only ($\mathbf{X}_t = \mathbf{X}_{t-1} + \Delta\mathbf{X}_t$) and passing raw observations as additional input to GNNs. We would like to investigate the effect of these interesting differences in future work.

## 5.3 ROAD TRAFFIC

Traffic speed forecasting on road networks is an important but challenging task, as the traffic dynamics exhibit complex spatiotemporal interactions. In this subsection, we demonstrate that R-SSM is comparable to the state-of-the-art GNN-based seq2seq baselines on a real-world traffic dataset. The METR-LA dataset (Li et al., 2018) contains 4 months of 1D traffic speed measurements that were recorded via 207 sensors and aggregated into 5 minutes windows. For this dataset, all conditional inputs $\mathcal{G} = (\mathcal{V}, \mathcal{E}, \mathbf{V}, \mathbf{E})$ and $\mathbf{U}_{1:T}$ are provided to R-SSM, in which $\mathcal{E}$ is constructed by connecting two sensors if their road network distance is below a threshold, $\mathbf{V}$ stores the geographic

---

[2]Data Source: STATS, copyright 2019.

positions and learnable embeddings of sensors, $\mathbf{E}$ stores the road network distances of edges, and $\mathbf{U}_{1:T}$ provides the time information (hour-of-day and day-of-week). We impute the missing values for training and exclude them from evaluation. GNF is not used because of GPU memory limitation. Following the settings in Li et al. (2018), we train our model on small time windows spanning 2 hours and use a 7:1:2 split for training, validation, and test.

The comparison of mean absolute forecast errors (MAE) is reported in Table 4. The three forecast horizons correspond to 15, 30, and 60 minutes. We give point predictions by taking the element-wise median of 2K Monte Carlo forecasts. Compared with DCRNN (Li et al., 2018) and GaAN (Zhang et al., 2018), R-SSM delivers comparable short-term forecasts but slightly worse long-term forecasts.

We argue that the results are admissible because: **(i)** By using MAE loss and scheduled sampling, the DCRNN and GaAN baselines are trained on the multi-step objective that they are later evaluated on, making them hard to beat. **(ii)** Some stochastic systems are inherently unpredictable beyond a few steps due to the process noise, e.g., the toy model in Section 5.1. In such case, multi-step MAE may not be a reasonable metric, and probabistic forecasts may be prefered. The average coverage probabilities (CP) of 90% prediction intervals reported in Table 4 indicate that R-SSM provides good uncertainty estimates. **(iii)** Improving the multi-step prediction ability of deep SSMs is still an open problem with a few recent attempts (Ke et al., 2019; Hafner et al., 2019). We would like to explore it in future work.

## 6 CONCLUSIONS

In this work, we present a deep hierarchical state-space model in which the state transitions of correlated objects are coordinated by graph neural networks. To effectively learn the model from observation data, we develop a structured posterior approximation and propose two auxiliary contrastive prediction tasks to help the learning. We further introduce the graph normalizing flow to enhance the expressiveness of the joint transition density and the posterior approximation. The experiments show that our model can outperform or match the state-of-the-arts on several time series modeling tasks. Directions for future work include testing the model on high-dimensional observations, extending the model to directly learn from visual data, and including discrete latent variables in the model.

### ACKNOWLEDGMENTS

This work was supported by the National Key Research and Development Program of China (No. 2018YFB0505000) and the Alibaba-Zhejiang University Joint Institute of Frontier Technologies. Fan Yang would like to thank Qingchen Yu for her helpful feedback on early drafts of this paper.

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

# A   APPENDIX

## A.1   MHA IMPLEMENTATION

Each attention head $k \in [K]$ is separately parameterized and operates as follows. For $\forall (j, i) \in \mathcal{E}$, it produces a message vector $\boldsymbol{\beta}_{j \to i}^k$ together with an unnormalized scalar weight $\omega_{j \to i}^k$. Then for $\forall i \in \mathcal{V}$, it aggregates all messages sent to $i$ in a permutation-invariant manner:

$$\left\{ \alpha_{p \to i}^k \right\}_{p \in \mathcal{N}_i^-} = \text{softmax} \left( \left\{ \omega_{p \to i}^k \right\}_{p \in \mathcal{N}_i^-} \right) , \quad \mathbf{a}_i^k = \sum_{p \in \mathcal{N}_i^-} \alpha_{p \to i}^k \boldsymbol{\beta}_{p \to i}^k .$$

Putting all $K$ heads together, it turns out that $\mathcal{M}_{j \to i} = \{(\omega_{j \to i}^k, \boldsymbol{\beta}_{j \to i}^k)\}_{k=1}^K$ and $\mathcal{A}_i = \{\mathbf{a}_i^k\}_{k=1}^K$. Specifically, each attention head is parameterized in a query-key-value style:

$$\forall i \in \mathcal{V} : \quad \tilde{\mathbf{v}}_i = \text{MLP}_v(\mathbf{v}_i) \in \mathbb{R}^{\tilde{d}_v} , \quad \tilde{\mathbf{h}}_i = [\mathbf{h}_i, \tilde{\mathbf{v}}_i, \mathbf{g}]$$

$$\mathbf{Q} = \tilde{\mathbf{H}} \mathbf{W}_Q , \quad \mathbf{A} = \tilde{\mathbf{H}} \mathbf{W}_A , \quad \mathbf{C} = \tilde{\mathbf{H}} \mathbf{W}_C$$

$$\boldsymbol{\beta}_{j \to i} = \mathbf{c}_j , \quad \omega_{j \to i} = \mathbf{q}_i^\top \mathbf{a}_j / \sqrt{d_q} + \text{MLP}_e(\mathbf{e}_{ji}) \in \mathbb{R}$$

for $\tilde{d} = d + \tilde{d}_v + d_g$, $\mathbf{W}_Q \in \mathbb{R}^{\tilde{d} \times d_q}$, $\mathbf{W}_A \in \mathbb{R}^{\tilde{d} \times d_q}$, and $\mathbf{W}_C \in \mathbb{R}^{\tilde{d} \times d_c}$.

## A.2   MODEL DETAILS

In this work, the READOUT function is implemented by passing the concatenation of the outputs of a *mean* aggregator and an element-wise *max* aggregator through a gated activation unit. The transition densities in the generative model are specified to be:

$$f_\theta^g \left( \mathbf{z}_t^g \big| \mathbf{h}_t^g \right) = \text{Normal} \left( \cdot \big| \boldsymbol{\mu}_\theta^g \left( \mathbf{h}_t^g \right), \boldsymbol{\Sigma}_\theta^g \left( \mathbf{h}_t^g \right) \right) , \tag{7}$$

$$f_\theta^\star \left( \mathbf{Z}_t \big| \mathbf{H}_t \right) = \prod_{i=1}^N \text{Normal} \left( \mathbf{z}_t^{(i)} \big| \boldsymbol{\mu}_\theta^\star \left( \mathbf{h}_t^{(i)} \right), \boldsymbol{\Sigma}_\theta^\star \left( \mathbf{h}_t^{(i)} \right) \right) , \tag{8}$$

where $\boldsymbol{\mu}_\theta^g$ and $\boldsymbol{\Sigma}_\theta^g$ (similarly $\boldsymbol{\mu}_\theta^\star$ and $\boldsymbol{\Sigma}_\theta^\star$) are 3-layer MLPs that share their first layer. $\boldsymbol{\Sigma}_\theta^g$ and $\boldsymbol{\Sigma}_\theta^\star$ output diagonal covariance matrices using the softplus activation. The proposal densities $r_\phi^g$ and $r_\phi^\star$ are specified in a similar way. Then GNFs can be stacked on top of $f_\theta^\star$ and $r_\phi^\star$ to make them more expressive.

## A.3   EXPERIMENTS

We use the Adam optimizer with an initial learning rate of 0.001 and a gradient clipping of 1.0 for all experiments. The learning rate was annealed according to a linear cosine decay. We set $\beta_1 = \beta_2 = 1.0$ for the auxiliary losses in all experiments.

**Synthetic toy dataset.**   A typical example in the dataset is visualized in Figure 2. The architectures of the models are specified as follows.

(a) VRNN: Using 128-dimensional latent variables and a two-layer, 512-unit GRU.

(b) GNN-AR: Using a two-layer GNN and an one-layer, 128-unit GRU shared by all nodes.

(c) R-SSM: We let $d_g = d_z = 8$. All RNNs are specified to be two-layer, 32-unit LSTMs. All MLPs use 64 hidden units. The generative model and the proposal both use a 4-head MHA layer. 4 SMC samples and a batch size of 16 are used in training.

**Basketball player movement.**   We let $d_g = d_z = 32$. All RNNs are specified to be two-layer, 64-unit LSTMs and all MLPs use 256 hidden units. The generative model uses one 8-head MHA layer and the proposal uses two 8-head MHA layers. Each GNF uses an additional MHA layer shared by the functions $s(\cdot)$ and $t(\cdot)$. 4 SMC samples and a batch size of 64 are used in training. Eight selected rollouts from the trained model are visualized in Figure 3.

**Road traffic.**   We let $d_g = d_z = 8$. A 32-dimensional embedding for each sensor is jointly learned as a part of the vertex attribute. All RNNs are specified to be two-layer, 32-unit LSTMs and all MLPs use 64 hidden units. The generative model and the proposal both use two 8-head MHA layers. 3 SMC samples and a batch size of 16 are used in training.

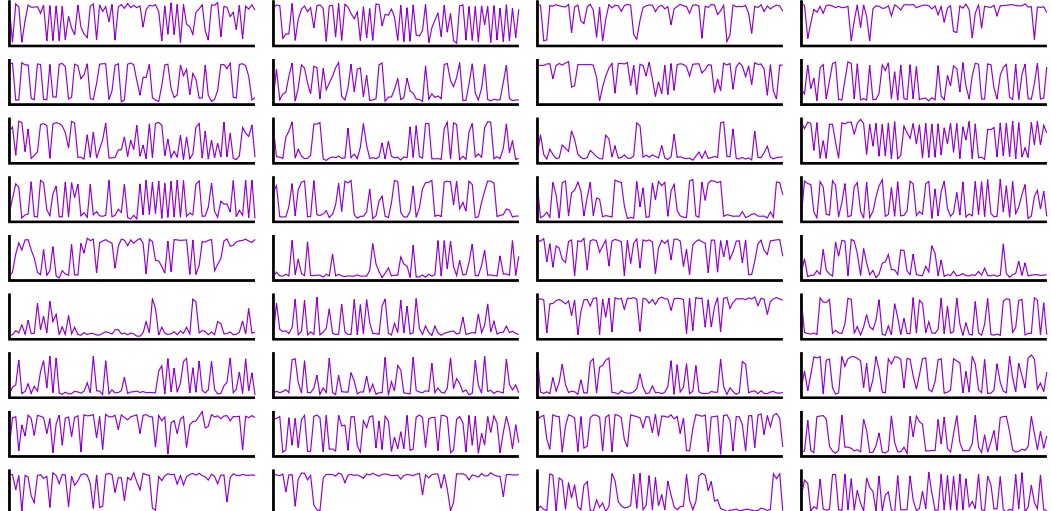

Figure 2: An example from the toy dataset ($N = 36, T = 80$).

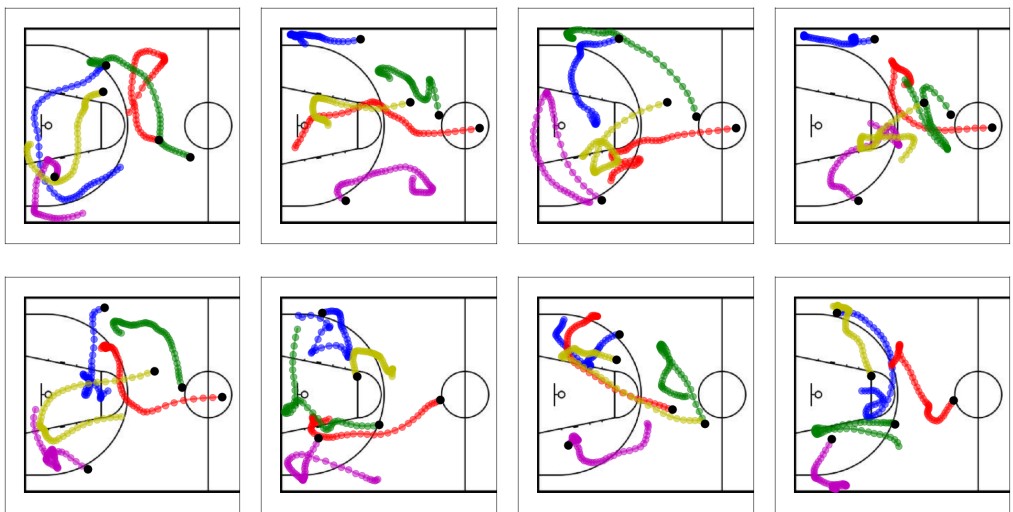

Figure 3: Selected rollouts from the trained model. Black dots represent the starting points.

### A.4 TRAINING

We optimize the VSMC bound estimated by the following SMC algorithm:

---

**Algorithm 1** Estimate the VSMC bound $\mathcal{L}_K^{\mathrm{SMC}}$

---

**Input:** graph $\mathcal{G}$, observations $\mathbf{X}_{1:T}$, exogenous inputs $\mathbf{U}_{1:T}$
**Require:** generative model $\{f_\theta^g, f_\theta^\star, g_\theta\}$, proposal $\{r_\phi^g, r_\phi^\star\}$, number of particles $K$

**for** $k = 1 \ldots K$ **do**
     Simulate $\mathbf{z}_{1,k}^g \sim r_\phi^g(\cdot | \mathcal{G}, \mathbf{X}_1, \mathbf{U}_1)$
     Simulate $\mathbf{Z}_{1,k} \sim r_\phi^\star(\cdot | \mathbf{z}_{1,k}^g, \mathcal{G}, \mathbf{X}_1, \mathbf{U}_1)$
     Set $w_1^k = \frac{f_\theta^g(\mathbf{z}_{1,k}^g) f_\theta^\star(\mathbf{Z}_{1,k} | \mathbf{z}_{1,k}^g, \mathcal{G}, \mathbf{U}_1) \prod_{i=1}^N g_\theta(\mathbf{x}_1^{(i)} | \mathbf{z}_{1,k}^{(i)}, \mathbf{z}_{1,k}^g, \ldots)}{r_\phi^g(\mathbf{z}_{1,k}^g | \ldots) r_\phi^\star(\mathbf{Z}_{1,k} | \ldots)}$
**end for**
Initialize $\hat{\mathcal{L}}_K^{\mathrm{SMC}} = \log \sum_{k=1}^K w_1^k / K$

**for** $t = 2 \ldots T$ **do**
     $\{\mathbf{z}_{<t,k}^g, \mathbf{Z}_{<t,k}\}_{k=1}^K = \mathrm{RESAMPLE}(\{\mathbf{z}_{<t,k}^g, \mathbf{Z}_{<t,k}, w_{t-1}^k\}_{k=1}^K)$
     **for** $k = 1 \ldots K$ **do**
         Simulate $\mathbf{z}_{t,k}^g \sim r_\phi^g(\cdot | \mathbf{z}_{<t,k}^g, \mathbf{Z}_{<t,k}, \mathcal{G}, \mathbf{X}_{\le t}, \mathbf{U}_{\le t})$
         Simulate $\mathbf{Z}_{t,k} \sim r_\phi^\star(\cdot | \mathbf{z}_{\le t,k}^g, \mathbf{Z}_{<t,k}, \mathcal{G}, \mathbf{X}_{\le t}, \mathbf{U}_{\le t})$
         Set $w_t^k = \frac{f_\theta^g(\mathbf{z}_{t,k}^g | \ldots) f_\theta^\star(\mathbf{Z}_{t,k} | \mathbf{z}_{\le t,k}^g, \mathbf{Z}_{<t,k}, \ldots) \prod_{i=1}^N g_\theta(\mathbf{x}_t^{(i)} | \mathbf{z}_{t,k}^{(i)}, \mathbf{z}_{\le t,k}^g, \mathbf{Z}_{<t,k}, \ldots)}{r_\phi^g(\mathbf{z}_{t,k}^g | \ldots) r_\phi^\star(\mathbf{Z}_{t,k} | \ldots)}$
         Set $\mathbf{z}_{\le t,k}^g = (\mathbf{z}_{<t,k}^g, \mathbf{z}_{t,k}^g)$, $\mathbf{Z}_{\le t,k} = (\mathbf{Z}_{<t,k}, \mathbf{Z}_{t,k})$
     **end for**
     Update $\hat{\mathcal{L}}_K^{\mathrm{SMC}} = \hat{\mathcal{L}}_K^{\mathrm{SMC}} + \log \sum_{k=1}^K w_t^k / K$
**end for**

**Output:** $\hat{\mathcal{L}}_K^{\mathrm{SMC}}$

---

In our model, dependencies on $\mathbf{z}_{<t,k}^g$ and $\mathbf{Z}_{<t,k}$ are provided through the compact RNN states $\mathbf{h}_{t,k}^g$ and $\mathbf{H}_{t,k}$. When GNFs are used in $f_\theta^\star$ and $r_\phi^\star$, density calculation and backpropagation are automatically handled by the TensorFlow Probability library.

To estimate the auxiliary objectives $\mathcal{L}_1^{\mathrm{aux}}$ and $\mathcal{L}_2^{\mathrm{aux}}$, we reuse the resampled unweighted particles $\{\{\mathbf{z}_{1:t,k}^g, \mathbf{Z}_{1:t,k}\}_{k=1}^K\}_{t=1}^{T-1}$ generated by Algorithm 1 to form Monte Carlo estimations for them:

$$\hat{\mathcal{L}}_1^{\mathrm{aux}} = \sum_{t=1}^{T-1} \sum_{i=1}^N \frac{1}{K} \log \frac{\lambda_{\psi,1}(\hat{\mathbf{z}}_{t,k}^{(i)}, \mathbf{c}_t^{(i)})}{\sum_{\mathbf{c} \in \Omega_{t,i}} \lambda_{\psi,1}(\hat{\mathbf{z}}_{t,k}^{(i)}, \mathbf{c})}, \quad \hat{\mathcal{L}}_2^{\mathrm{aux}} = \sum_{t=1}^{T-1} \sum_{i=1}^N \frac{1}{K} \log \frac{\lambda_{\psi,2}(\hat{\mathbf{h}}_{t,k}^{(i)}, \mathbf{c}_t^{(i)})}{\sum_{\mathbf{c} \in \Omega_{t,i}} \lambda_{\psi,2}(\hat{\mathbf{h}}_{t,k}^{(i)}, \mathbf{c})}$$

where $\hat{\mathbf{z}}_{t,k}^{(i)} = [\mathbf{z}_{t,k}^g, \mathbf{z}_{t,k}^{(i)}]$, $\hat{\mathbf{h}}_{t,k}^{(i)} = \mathrm{MLP}_\psi(\sum_{j \neq i \,\wedge\, j \in \mathcal{N}_i^-} \mathbf{h}_{t,k}^{(j)})$, and $\Omega_{t,i}$ is a set that contains $\mathbf{c}_t^{(i)}$ and some negative samples selected from the future summaries of other vertices within the minibatch.

## A.5 GRAPH NORMALIZING FLOW

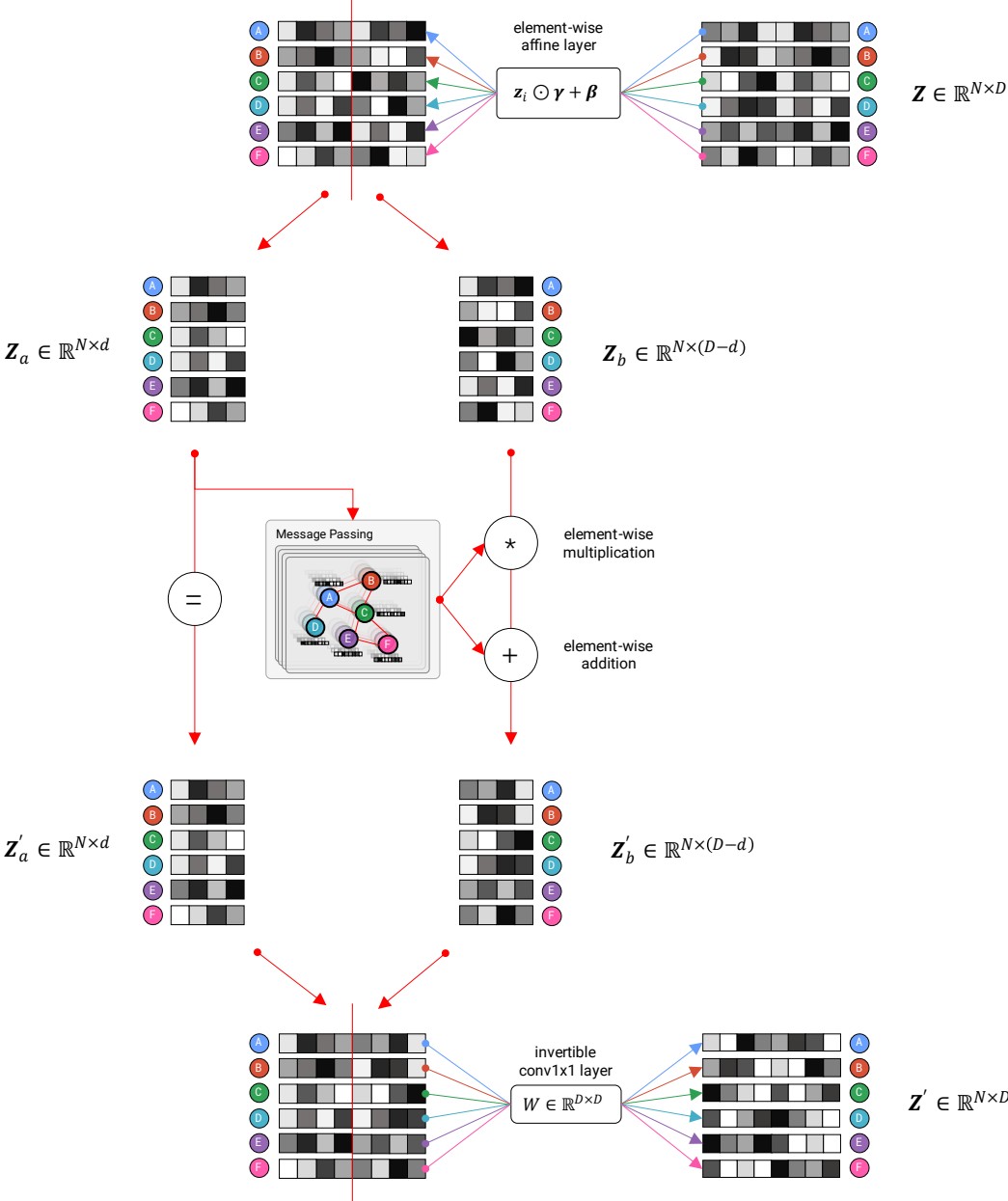

Figure 4: Visual illustration of a GNF. Multiple GNFs can be stacked together to achieve more expressive transformation.

The element-wise affine layer is proposed by Kingma & Dhariwal (2018) for normalizing the activations. Its parameters $\gamma \in \mathbb{R}^D$ and $\beta \in \mathbb{R}^D$ are initialized such that the per-channel activations have roughly zero mean and unit variance at the beginning of training. The invertible linear transformation $\mathbf{W} \in \mathbb{R}^{D \times D}$ is parameterized using a QR decomposition (Hoogeboom et al., 2019).

