# OpenReview forum: "Relational State-Space Model for Stochastic Multi-Object Systems"
_ICLR.cc/2020/Conference — Accept (Poster)_

### Official Review · AnonReviewer2 · 2019-10-24
**Official Blind Review #2**

**Rating:** 6

**Review:**

Summary of what the paper claims and contributes
---
This paper proposes a hierarchical latent variable model of sequential dynamic processes of multiple objects when each object exhibit significant stochasticity. The model leverages Graph Neural Networks to design the architecture of the model. Normalizing flows are used to construct "Graph Normalizing Flows" that model transition densities of the per-object latent random variables, although the full model has more latent variables than tabouthe per-object latents. The approach applies and combines a host of methods to design the objective, the model, and optimize the model under the objective -- for the objective: variational inference, Contrastive Predictive Coding; for the model: normalizing flows, graph neural networks (and their combination, termed by the authors as Graph Normalizing Flows); for the optimization: variational sequential Monte Carlo. From my understanding, the main claims are that the model is a flexible way to incorporate relational information, and that the resulting model outperforms other multi-object time-series forecasting approaches.

Experiments are conducted on a synthetic toy dataset of 1d observations per-object, a real dataset of basketball player movement, corresponding to 2d observations of each object's position (the agent's and the ball's positions), and a real dataset of traffic speed forecasting (1d observations per-object). The experiments demonstrate that the model compares favorably to a baseline and a few other time-series forecasting models from the literature. The experiments show that the method achieves higher test log-likelihood (lower-bounds), and lower MSE than the compared methods. The paper discussed many related works, but it's not clear why the specific methods were chosen for comparisons. More motivation is needed for these comparisons.

Evaluation
---

>Originality:
Are the tasks or methods new?
The method is new.

Is the work a novel combination of well-known techniques?
Yes, albeit a lot of techniques.

Is it clear how this work differs from previous contributions?
Besides the fact that the proposed method is a combination of many techniques, the main / key technical difference was actually unclear to me.

Is related work adequately cited?
From my understanding, yes, mostly; there's just a few places where the paper could be situated better:
- Sec 1: "as they have been shown to be fundamental NN building blocks ...". Citations needed (e.g. include the Graph NN citations later here)
- Sec 1: "R-SSM achieves superior test likelihood and good prediction performance". When compared to what methods (citations to VRNN and GNN-AR/NRI needed here too)?
- The paper contrasted the work to other dynamics modeling works with the statement "R-SSM differs from all these works by introducing structured latent variables to represent the uncertainty on state transition and estimation". The paper also said VRNNs were the most related, because they employ "hidden states of per-agent VRNNs interact through GNNs". The paper (PRECOG, arXiv:1905.01296) appears related as it also constructs a latent variable graphical model of stochastically-transitioning multiple interacting objects with normalizing flows. Like the authors said of Graph VRNNs, it appears that one of the biggest differences to PRECOG looks like the proposed method includes more sources of stochasticity outside of the objects (the global uncertainty) to model the observations.

>Quality:
Is the submission technically sound?
Yes, although not much motivation is given for many of the specific design decisions adopted. The model is a complicated combination of many techniques from the literature, which makes it difficult to understand the importance of each specific piece of the model.
- For instance, why was global stochasticity introduced? Because per-object stochasticity is modeled with normalizing flows, the model would have been trainable with maximum likelihood estimation of a distribution over observations that's efficiently analytically computable, but after the addition of global stochasticity, the joint distribution over the random variables becomes intractable, and training must resort to variational inference. This leads to, as the paper mentioned, RSSM suffering from posterior collapse, which, to my knowledge, does not occur when training direct maximum likelihood estimation procedures (e.g. GLOW, RealNVP).
- As mentioned above, it's not clear what the key technical innovation of this approach is (Graph Normalizing Flows?) It would be good if the claimed innovation was made clear in both the abstract and the introduction.

Are claims well supported by theoretical analysis or experimental results?
The experimental results provide evidence that 1) the model can flexibly incorporate relation information and some evidence that 2) the model outperforms some multi-object time-series forecasting approaches from the literature.

Is this a complete piece of work or work in progress?
The submission appears complete.

Are the authors careful and honest about evaluating both the strengths and weaknesses of their work?
Yes, and they included discussion of when the model is under-performing w.r.t. other methods (at longer time horizons).

>Clarity:
Is the submission clearly written?
As mentioned above, many of the numerous design decisions are not motivated well. Essentially, the paper is not very self-contained.

Is it well organized?
Yes.

Does it adequately inform the reader?
No -- the paper assumes a lot of prior knowledge, which most potential readers would likely possess. This issue would be less significant if the main innovation was clearer.

>Significance:
Are the results important?
It's difficult to tell, because it's not very clear why, of the many cited works, the specific ones chosen for comparison were used. More motivation for the comparison is needed.

Are others (researchers or practitioners) likely to use the ideas or build on them?
Possibly, although the model is complicated, which makes it less likely.

Does the submission address a difficult task in a better way than previous work?
It's plausible that the model performs better, but it's unclear how widely-applicable the method is. The model appears to be restricted to lower dimensional settings.

Does it advance the state of the art in a demonstrable way?
It's plausible, but I am not sure, as mentioned above.

Does it provide unique data, unique conclusions about existing data, or a unique theoretical or experimental approach?
No

Additional feedback
---
Sec 5.2 describe what "OOB rate of the rollouts" means.

Sec 3.3 Coupling layer equation is missing a final parenthesis.

3.3 "complicate the learning a lot." -> "significantly complicate the learning." (original is subjectively too informal)

Sec 5.2 "like to dig into it in future work" -> "like to explore it in future work" (original is subjectively too informal)

Recommendation
---
It is unclear what the main key contributions are, outside of usually performing better than some related methods. This lack of clarity, along with a complicated design that is not very well-motivated, make me lean towards rejection. An improved version of this paper would make clear what the precise contributions are, explain how the related work did not achieve them, and why the specific comparisons were adopted.


**Experience Assessment:**

I have read many papers in this area.

**Review Assessment: Checking Correctness Of Derivations And Theory:**

N/A

**Review Assessment: Checking Correctness Of Experiments:**

I assessed the sensibility of the experiments.

**Review Assessment: Thoroughness In Paper Reading:**

I read the paper at least twice and used my best judgement in assessing the paper.

---

> ### Author Response · Authors · 2019-11-12
> **Response to Reviewer #2: Part 1**
>
> Thank you for the detailed reviews and constructive feedback. Below, we summarize your questions and address them in order.
>
> 1. RE: Key contributions
> Our key contributions are twofold:
> (1) We propose R-SSM, a new GNN-backboned sequential LVM for stochastic multi-object systems, moving beyond relatively simple deterministic multi-object dynamics models. This contribution distinguishes our work from GNN-based AR models.
> (2) We introduce GNF to construct expressive joint distributions for random variables indexed by graph nodes, which helps enhance the capability of R-SSM. This contribution breaks the commonly adopted independence assumption and makes our work distinct from straightforward combinations of GNN and deep SSM (e.g., GVRNN).
>
> 2. RE: Baseline selection
> As our model is a GNN-based sequential LVM, we regard GNN-based AR models and sequential LVMs for multi-agent systems as our main competitors. This leads to the GNN-AR/DCRNN/MI-VRNN baselines in the original submission and the newly added GVRNN baseline in the revised submission.
>
> 3. RE: Connections to PRECOG [1]
> Thanks for bringing this work to our attention. In the latest revision, we cited it in the Related Work section (under the "GNN-based dynamics modeling" subsection).  We notice that a large part of your concern is caused by comparing R-SSM with the ESP model proposed in [1]. From our viewpoint, the main connections between them are: (a) They both target the multi-agent trajectory modeling task; (b) ESP includes a neural message passing module (the PastRNN component in Table 3 of [1]), which can be regarded as a GNN. Except for the two connections, ESP is significantly different from R-SSM as stated in the following comments.
> (1) R-SSM is a latent variable model (LVM), while ESP is a normalizing flow model that can be learned by maximizing the exact likelihood. Calling ESP a "latent variable model" is not accurate in our opinion (although [1] uses this term), as LVMs (e.g., VAE) generally model the joint distribution $p(x, z) = p(x|z)p(z)$ and are learned by approximate inference.
> (2) A more subtle fact is that ESP is also an **AR** model because it employs a specially designed one-layer autoregressive flow. At each time step, this AR model outputs a block-diagonal Gaussian density, of which mean and covariance are parameterized as a function of the history. The reason why SSMs may be more expressive than AR models (at least in theory) still makes sense here. Specifically, the AR interpretation of ESP only assumes stochasticity in the observation model,  while GVRNN and R-SSM assume stochasticity in both the transition model and the observation model.
> (3) We admit that ESP is more related to GNF, as they both are normalizing flows for modeling multi-agent data. However, we emphasize that GNF is used for modeling the joint distribution of per-agent latent states at each transition step of R-SSM, while ESP directly models multi-agent trajectories. We are aware that GNF by itself can be applied to model multi-agent trajectories, but this is not the concern of our current work.
>
> 4. RE: "Because per-object stochasticity is modeled with normalizing flows, the model would have been trainable with maximum likelihood estimation of a distribution over observations that's efficiently analytically computable, but after the addition of global stochasticity, the joint distribution over the random variables becomes intractable, and training must resort to variational inference"
> (1) This is a significant misunderstanding of our work and we would like to clarify. Even without the global latent process, our model is still a deep SSM (similar to GVRNN) with intractable data likelihood $p(X_{1:T}) = \int p(X_{1:T}, Z_{1:T}) dZ_{1:T}$, and training algorithms based on approximate inference must be adopted. As stated above, when GNF is used in R-SSM, it models the stochasticity in joint state transitions rather than directly models the observations. So, in our framework, GNF is just an enhancement for R-SSM, and the use of it does not change the fact that R-SSM requires approximate inference (again, even without the global latent process).
> (2) Now, we answer the question why the global latent process is introduced. Using the global latent process is motivated by the fact that the Graph Network [2] keeps track of a graph-level global attribute. R-SSM can be viewed as a natural generalization of recurrent node-centric GN to stochastic settings. We believe that the global latent state helps encode global information and uncertainty that are shared by all nodes. For example, when using R-SSM to model the trajectories of offensive basketball players only, the global state may help encode the uncertainty on the effect of the defensive team. We would like to explore the representations encoded in global and local latent states in our future work.

---

> > ### Author Response · Authors · 2019-11-12
> > **Response to Reviewer #2: Part 2**
> >
> > (3) Finally, the idea that designing normalizing flows to directly model multi-agent trajectories is interesting. For instance, if the dimensionality of per-agent per-step observation is $\geq$ 2, a natural idea is to combine ESP and GNF by replacing Equation (3) of [1] with GNF.
> >
> > 5. RE: Complicated design is not very well-motivated
> > We hope the above response clarifies the motivation of our design choices. As for the model complexity, we would argue that the base R-SSM model is somehow simpler than GVRNN (differences are pointed out in the last paragraph of revised Section 5.2). GNF is relatively easy to implement with the help of TFP library. We are sorry about the lack of explanation for how several parts of our model are connected in the original submission. In the revised paper, we added a new subsection in Appendix A.4, which includes a sketch of the SMC algorithm and shows how the learning objectives are estimated. We also include a visual illustration of GNF in Appendix A.5. We hope these revisions clarify how the generative model, the proposal, GNF, and the auxiliary model are jointly learned.
> >
> > 6. Minor points
> > > what "OOB rate of the rollouts" means.
> > It means the percentage of time steps when some player is out-of-bounds.
> > > Coupling layer equation is missing a final parenthesis.
> > This is not true.
> > > ... original is subjectively too informal
> > > "fundamental NN building blocks ...". Citations needed.
> > We adopt your suggestions in the revised paper. Thanks.
> >
> >
> > We hope our response resolves your main concerns and would be grateful if you could increase the rating.
> >
> > [1] Nicholas Rhinehart, et al. PRECOG: Prediction Conditioned On Goals in Visual Multi-Agent Settings. ICCV 2019 [Arxiv 1905.01296]
> > [2] Peter W Battaglia, et al. Relational inductive biases, deep learning, and graph networks. Arxiv 2018

---

> > > ### Comment · AnonReviewer2 · 2019-11-14
> > > **Response to "Response to Reviewer #2"**
> > >
> > > Thank you for responding to my concerns and clarifying your approach. I'm going to increase my rating, however, I think that the contributions and experiments introduction paragraphs need to be improved in order to further clarify 1) the exact contributions in a self-contained way 2) how the experiments validate these claims of contributions. I'm assuming these writing issues can be addressed before the rebuttal period closes.
> > >
> > > Quoting from the paper: "(i) We propose a new deep sequential hierarchical LVM, termed as the relational state-space model...providing an alternative way to model time series data observed from multi-object systems."
> > > Alternative to what? It would be clearer to explicitly say the exact properties of the approach that are novel.
> > >
> > > Quoting from the paper: "(ii) We suggest using the graph normalizing flow (GNF) ...  which enables R-SSM to better capture the joint evolutions of correlated stochastic subsystems"
> > > Better than what? It would be clearer to rephrase this contribution with respect to prior work. The contributions paragraph is best when it's self-contained.
> > >
> > > The experiments section would be clearer if its very first paragraph stated what the experiments in Sec 5.1 and Sec 5.2 are going to investigate, and relate that to the specific baselines chosen. Ideally, this explanation should tie directly into (at least some of) the three listed contributions in the introduction. Currently, the baselines are described after the toy task, which makes the reader have to dig for _why_ the specific baselines are chosen.
> > >
> > > Finally, in your response: "moving beyond relatively simple deterministic multi-object dynamics models. This contribution distinguishes our work from GNN-based AR models.", and in the paper "AR models based on recurrent (G)NNs can be viewed as deterministic variants of SSMs"
> > > This terminology is confusing, because it sounds like you're claiming AR models always lead to deterministic predictions, whereas some AR models generate stochastic predictions (e.g. IAF and MAF). My understanding is that it's been used interchangeably also in the literature; regardless, please clarify and either stick to a single definition of AR, or consider using a different word, or qualify it with "deterministic autoregressive" or "stochastic autoregressive" where it's used in the paper. When "deterministic" and "stochastic" are used, make clear what quantity is deterministic or stochastic, rather than calling it, e.g. "deterministic variant".

---

> > > > ### Author Response · Authors · 2019-11-14
> > > > **Paper Revised**
> > > >
> > > > Thank you very much for raising your rating! This means a lot to us.
> > > >
> > > > We sincerely appreciate your feedback. We admit that these paragraphs are a bit confusing and need to be improved. In the latest revision, we reworked the summary for our contributions and added an overview of the experiment setup at the beginning of Section 5. The confusing terminology "deterministic variants" was also been replaced with a clearer description, and a proper citation was added to make this statement more accessible. We hope that these modifications make the paper clearer and address your remaining concerns.
> > > >
> > > > Besides, we would like to reply to your comment on AR models and stochastic predictions. In the original submission, by "deterministic variants of SSMs" we mean that the state transition model is restricted to being deterministic, rather than the observation model. We are not claiming that AR models always lead to deterministic predictions, as every AR model can generate stochastic predictions (not only autoregressive flows). AR models are an important family of probabilistic generative models that make use of the very general chain rule of probability to factorize distributions, i.e., $p(x_{1:T}) = \prod_{t=1}^{T}{p(x_t | x_{<t})}$. The whole point is that when $p(x_t | x_{<t})$ is designed to be simple (e.g., unimodal distribution or its finite mixture), then given a fixed history $x_{<t}$, the possibility for the future $x_t$ will be restricted. This is why introducing latent variables may help. Thank you for pointing out the confusion, and we hope the revision has make it clearer.

---

### Official Review · AnonReviewer1 · 2019-10-28
**Official Blind Review #1**

**Rating:** 6

**Review:**

* Summary:
The paper presents a relational state-space model that simulates the joint state transitions of correlated objects which are hierarchically coordinated in a graph structure. A structured posterior approximation is developed based on sequential graph neural networks. Two auxiliary contrastive predictive losses are proposed to help circumvent the posterior collapse problem. Graph normalizing flow is further incorporated into the framework to make the joint state transition density more expressive. The proposed R-SSM shows performance gains over state-of-the-arts in three benchmarks.

* Comments:
The paper is generally well written and technically sound. The framework, including formulation of each of its components, is well defined. It is also helpful that the authors included preliminaries of the literature. The number of experiments are adequate. However, there are a few parts that require more extensive clarification and analysis.
1. Different parts of Section 3 appear to be rather disconnected, the reader still has a hard time figuring out how the learning of the whole framework is carried out. It is desirable to include a sketch of learning algorithm.
2. In the formulation GNF, what is the intuition or principle to decouple the state Z_t into two parts Z_a and Z_b? How does the mapping of Z_b into Z'_b help to make the state transition distribution more expressive?
3. In Section 5.3, the authors mention that GNF was not used due to memory cost. Could it be discussed more thoroughly about the complexities of learning R-SSM and GNF?
4. The model keeps track of a global state z^g, but it is not analyzed in experiments. It is strongly recommended that the authors discuss about the (global and individual) states and their transitions. It would provide great insights on how multiple objects interact with each other.

Minor point:
1. Please clarify what is X_{t-h} in Table 3?

**Experience Assessment:**

I have read many papers in this area.

**Review Assessment: Checking Correctness Of Derivations And Theory:**

I assessed the sensibility of the derivations and theory.

**Review Assessment: Checking Correctness Of Experiments:**

I assessed the sensibility of the experiments.

**Review Assessment: Thoroughness In Paper Reading:**

I read the paper at least twice and used my best judgement in assessing the paper.

---

> ### Author Response · Authors · 2019-11-11
> **Response to Reviewer #1**
>
> Response to Reviewer #1
>
> Thank you for the supportive reviews and valuable feedback! We are glad that you are positive about our paper. Below, we address your concerns and questions in order.
>
> 1. RE: Different parts of Section 3 appear to be rather disconnected.
> Thanks for pointing it out. Following your suggestion, we added a new subsection in Appendix A.4 of the revised paper, which includes a sketch of the SMC algorithm and shows how the VSMC bound and auxiliary objectives are estimated under the SMC framework. We hope it clarifies how the generative model, the proposal, GNF, and the auxiliary model are jointly learned.
>
> 2.1 RE: The intuition or principle to partition the variable in the coupling layer
> The authors of RealNVP [1] introduced the idea that partitions the multidimensional random variable $Z$ into two disjoint parts $Z_a$ and $Z_b$, keeps one part $Z_a$ unchanged (as $Z_a^{'}$), and reversibly transforms the other part $Z_b$ into $Z_b^{'}$ using an invertible function $f$ parameterized by the unchanged part $Z_a$. With such a design, the overall transformation is ensured to be reversible as follows: Given $Z'$, we can recover $Z_a$ by copying $Z_a^{'}$. Then we know the function $f$ and thus its inverse $f^{-1}$, with which $Z_b^{'}$ can be mapped back into $Z_b$. GNF is also designed under this general framework.
>
> 2.2 RE: How GNF make the state transition distribution more expressive?
> We first answer the question that how a general coupling layer helps transform a diagonal Gaussian into a more expressive distribution. Diagonal Gaussian distribution assumes that different dimensions of the random variable $Z$ are independent. Adding coupling layers on top of it introduce dependencies among the dimensions, e.g., $Z_b^{'}$ can depend on $Z_a^{'}$. This enables the resulting distribution to model data with complex interdimensional dependencies. Now we answer the original question. To help understand the basic idea of GNF, a visual illustration of the GNF architecture is provided in Appendix A.5 of the revised paper. With the message passing operation inside the coupling layer, GNF introduces inter-node dependencies and thus make the resulting distribution have the potential to model correlated node states.
>
> 3 RE: The complexities of learning R-SSM and GNF
> Training R-SSM is generally time-consuming due to the sequential dependency (similar to RNN training). On a single GTX1080Ti GPU, training the base R-SSM model for the basketball gameplay dataset requires 3-4 days to converge. Adding 4-8 GNFs will slow down the training (~10 days) because: (1) The additional computation inside GNFs will be performed at each time step; (2) The memory cost limits us to small batch sizes. In theory, the latter problem can be resolved because the intermediate activations of GNFs need not be stored (can be recomputed inversely during backpropagation). However, we currently rely on the TensorFlow Probability library, which does not support this feature yet.
>
> 4. RE: Global state $z^g$
> Using the global latent process is motivated by the fact that the Graph Network [2] keeps track of a graph-level global attribute. R-SSM can be viewed as a natural generalization of recurrent node-centric GN to stochastic settings. We believe that the global latent state helps encode global information and uncertainty that are shared by all nodes. For example, when using R-SSM to model the trajectories of offensive basketball players only, the global state may help encode the uncertainty on the effect of the defensive team. We would like to explore the representations encoded in global and local latent states in our future work.
>
> 5. RE: The $X_{t-h}$ baseline
> We are sorry about the lack of explanation for this baseline in the original submission. In the revised paper, we clarify it in the caption of Table 4 (previous Table 3). It simply outputs $X_{t-h}$ to predict $X_t$.
>
> We hope our response resolves your main concerns and would be grateful if you could change the rating to "Accept".
>
> [1] Laurent Dinh, Jascha Sohl-Dickstein, and Samy Bengio. Density Estimation using Real NVP. ICLR 2017
> [2] Peter W Battaglia, et al. Relational inductive biases, deep learning, and graph networks. Arxiv 2018

---

### Official Review · AnonReviewer3 · 2019-11-07
**Official Blind Review #3**

**Rating:** 3

**Review:**

This manuscript proposes a novel approach to dynamic modeling of time series data based on a state space model that incorporates a Graph Neural Network to model the relationship between the dynamics of different objects. The (complex) architecture is described in details and to some extent justified before its predictive performance is demonstrated on several simulated and real datasets.
Overall, while the authors provide evidence of a better predictive performance with respect to several baselines, I am left a bit unconvinced by the study for the following reasons:
1.	Lack of relevant baselines: it seems clear that the overall purpose of the approach and the nature of the dataset require fitting a state space model, however, baseline are overall focused on recurrent and autoregressive models which seem underequipped to address these problems. I wonder if the choice of more relevant state space model baseline would convincingly show a true benefit of the proposed approach. In particular, there is likely a large number of variations of the Kalman filter and particle filters that might be relevant. For example, the ensemble Kalman filter has proved accurate for weather forecasting.
2.	Lack of interpretability: given the proposed approach relies on an intuitive representation of the model as representing the dynamics of several object tied by an interaction graph, the purely predictive results are not really matching the expectation of the reader to “see” how well these interactions are captured by the model. Can we check in some way that the latent graphical model is learnt properly, even in a toy dataset?
3.	Writing of the methods section: This is a more vague comment, but while reading the overall description of the approach, one is left wondering how critical are each part of the model and whether some complexity could be spared. Moreover, some descriptions are difficult to follow, perhaps for the reader less familiar with customary design choices in dynamic neural networks, e.g. the objective in section 3.4. In addition, some statements seem at least to lack justification, e.g. stating that AR approaches lead to unimodal distributions at the bottom of page 2.


**Experience Assessment:**

I do not know much about this area.

**Review Assessment: Checking Correctness Of Derivations And Theory:**

I assessed the sensibility of the derivations and theory.

**Review Assessment: Checking Correctness Of Experiments:**

I assessed the sensibility of the experiments.

**Review Assessment: Thoroughness In Paper Reading:**

I read the paper thoroughly.

---

> ### Author Response · Authors · 2019-11-13
> **Response to Reviewer #3**
>
> Thanks for your constructive feedback. We address your concerns in order.
>
> 1. RE: Lack of relevant baselines.
> (1) >"baselines are overall focused on recurrent and autoregressive models which seem underequipped to address these problems"
> First of all, we clarify that this statement is not accurate because stochastic RNNs (e.g., VRNN and its variants) are not AR models; rather, they are sequential latent variable models (LVMs). A subtle fact is that all stochastic RNNs can be viewed as SSMs parameterized by neural networks. This was detailedly discussed by Marco Fraccaro in his PhD thesis [1] (Section 4.3 - 4.5). The basic idea is that the recurrent connections in stochastic RNNs can be viewed as the exogenous input in traditional SSM formulation, and the latent variables correspond to SSM states. So stochastic RNNs are also be referred to as deep SSMs [1] (Section 4.3). Deep SSMs generally outperform traditional SSMs (e.g., Linear Dynamical Systems), and thus we do compare with strong SSM baselines. In addition, for the basketball gameplay dataset, we add a new strong baseline (GVRNN) in Section 5.2 of the revised paper and discuss the differences in design choices. Both MI-VRNN and GVRNN are state-of-the-art baselines on this dataset.
> (2) KF/EKF/PF
> They are inference algorithms, not models. To use one of them, a generative model must be specified at first (LDS for KF, nonlinear non-Gaussian SSM for EKF/PF). If the model contains unknown parameters, these methods can be combined with MLE or MCMC to learn the parameters. In our initial experiment on the toy dataset, we did try fitting an LDS for each example by MLE (using two R packages: "dlm" and "KFAS"), but it turns out that the results are not very stable and generally worse than VAR.
>
> 2. RE: Lack of interpretability
> We agree that currently interpreting a learned R-SSM is not easy, as the model is parameterized with NNs and contains multidimensional continuous latent states. We are aware that recent techniques developed for explaining GNNs [2] may help. As for the model design, discrete latent variables may be a better choice if we care about interpretability. We would like to explore these directions in our future work.
>
> 3. RE: Writing of the methods section
> Thank you for pointing it out. We are sorry about the lack of explanation in the original submission for how several parts of our model are connected. In the revised paper, we added a new subsection in Appendix A.4 (page 17), which includes a sketch of the SMC algorithm and shows how the learning objectives are estimated. We also include a visual illustration of GNF in Appendix A.5 (page 18). We hope these revisions clarify how the generative model, the proposal, GNF, and the auxiliary model are jointly learned.
> > some statements seem at least to lack justification, e.g. stating that AR approaches lead to unimodal distributions at the bottom of page 2.
> When using AR models for continuous observations, it is common to specify the conditional distribution $p(x_t | x_{<t})$ to be a Gaussian (which is unimodal) or a finite mixture of Gaussians for simplicity and tractability. For discrete observations, the categorical distribution can be viewed as a finite mixture of point masses. Integrating out the continuous latent states in SSM leads to an infinite mixture of the observation density, which may be more expressive.
>
> We hope our response resolves your main concerns and would be grateful if you could increase the rating.
>
> [1] Marco Fraccaro. Deep Latent Variable Models for Sequential Data. PhD thesis, 2018.
> [2] Rex Ying, et al. GNNExplainer: Generating Explanations for Graph Neural Networks. NeurIPS 2019

---

### Author Response · Authors · 2019-11-11
**Paper Update Overview**

First of all, we sincerely thank the reviewers for their time and appreciate all the detailed reviews and constructive feedback. The revised paper has been uploaded to address some of their concerns and questions. Our modification is summarized below, and the comments of reviewers will be replied individually. We sincerely hope reviewers revisit the score in light of our revision and response.

Major revisions:

1. Training details: Following the suggestion by Reviewer #1, in order to make it more clear how different parts of Section 3 are connected, a new subsection is added in Appendix A.4 (page 17) to describe the learning procedure. We show how the VSMC bound and auxiliary objectives are estimated under the SMC framework.

2. GNF: A visual illustration of the GNF architecture is provided in Appendix A.5 (page 18). We hope that it makes the idea of GNF more accessible to the reviewers and general readers. We also discuss the parameterization of the element-wise affine layer and the invertible conv1x1 layer in there to make it clearer how GNFs are initialized in training.

3. New baseline: New experiment results obtained after submission is reported in newly added Table 3 (page 9), in which we show test log-likelihood comparison with the Graph VRNN (GVRNN) model [1] mentioned in the Related Work section of the original submission. The setting in [1] differs from the original MI-VRNN baseline [2] by additionally modeling the trajectory of the basketball. As the implementation of [1] is not publicly available, we have contacted the authors to make sure that the same preprocessing procedure is applied to the dataset and, therefore, the test log-likelihoods are directly comparable. We find that the ELBO of R-SSM is higher than GRNN and VRNN but is lower than GVRNN. The VSMC bound for R-SSM is closer to the ELBO of GVRNN. In the newly added last paragraph of Section 5.2 (page 9), We point out different design choices made for GVRNN and R-SSM (besides the global latent process of R-SSM), which may be the reason why GVRNN outperforms R-SSM and are worth investigating in our future work.

4. Writing issues: In response to the suggestions by Reviewer #2, we rework the summary for our contributions in the Introduction section and summarize the experiment setup at the beginning of Section 5.

Minor revisions:

1. In response to the feedback by Reviewer #1, we clarify the baseline $\mathbf{X}_{t-h}$ in the caption of Table 4.
2. In response to the feedback by Reviewer #2, we cite [3] in the Introduction section and make our wording less informal.
3. New citations: We cite PRECOG [4] (as suggested by Reviewer #2) and MFP [5] (recently uploaded to arXiv) in the Related Work section.


[1] Raymond A. Yeh, et al. Diverse Generation for Multi-Agent Sports Games. CVPR 2019
[2] Eric Zhan, et al. Generating Multi-Agent Trajectories using Programmatic Weak Supervision. ICLR 2019
[3] Peter W Battaglia, et al. Relational inductive biases, deep learning, and graph networks. Arxiv 2018
[4]  Nicholas Rhinehart, et al. PRECOG: Prediction Conditioned On Goals in Visual Multi-Agent Settings. ICCV 2019 [arXiv 1905.01296]
[5] Yichuan C Tang and Ruslan Salakhutdinov. Multiple Futures Prediction. NeurIPS 2019 [arXiv 1911.00997]

---

### Decision · Program_Chairs · 2019-12-19

**Decision:**

Accept (Poster)

**Comment:**

The paper proposed what is termed Relational State Space Model (R-SSM) that can be used for modeling interacting time-series data. The model essentially consists of a set of (nonlinear) state space models whose states are jointly evolved in a way that take into account a known interaction structure between them (the relational part, even though technically it is just a coupling structure -- the term relational structure in the past has been used for models with objects and classes, for example see the difference between "coupled HMM" vs "relational HMM"). The authors also proposed a graph normalizing flow operation to model the joint state evolution. The main weakness of the paper is in the complexity of the model. However, from a modeling point of view, R-SSM seems suitable in situation when the interaction structure is known, and this is demonstrated in the experimental results when comparing against the baselines.